# Prioritized and predictive intelligence of things enabled waste management model in smart and sustainable environment

**Sushruta Mishra[1], Lambodar Jena[2], Hrudaya Kumar Tripathy[1], Tarek Gaber[3,4]\***

**1** School of Computer Engineering, Kalinga Institute of Industrial Technology Deemed to be University, Bhubaneswar, India, **2** Department of Computer Science and Engineering, Koneru Lakshmaiah Education Foundation, Vaddeswaram, A.P., India, **3** School of Science, Engineering and Environment, University of Salford, Salford, United Kingdom, **4** Faculty of Computers and Informatics, Suez Canal University, Ismailia, Egypt

\* t.m.a.gaber@salford.ac.uk

**Data Availability Statement:** All relevant data are within the manuscript and its Supporting Information files.

## Abstract

Collaborative modelling of the Internet of Things (IoT) with Artificial Intelligence (AI) has merged into the Intelligence of Things concept. This recent trend enables sensors to track required parameters and store accumulated data in cloud storage, which can be further utilized by AI based predictive models for automatic decision making. In a smart and sustainable environment, effective waste management is a concern. Poor regulation of waste in surrounding areas leads to rapid spread of contagious disease risks. Traditional waste object management requires more working staff, increases effort, consumes time and is relatively ineffective. In this research, an Intelligence of Things Enabled Smart Waste Management (IoT-SWM) model with predictive capabilities is developed. Here, local sinks (LS) are deployed in specified locations. At every instant, the current status of smart bins in each LS is notified to users to determine the priority level of LS to be emptied. Based on aggregated sensor values for the three smart bins, LS weight and poison gas value, the priority order of emptying LS is computed, and decision is made whether to notify the users with an alert message or not. It also helps in predicting the LS, which is likely to be filled up at a faster rate based on assigned timestamp. This model is implemented in real time with many LS and it was observed that bins, which were close to more crowded sites filled up faster compared to sparse populated areas. Random forest algorithm was used to predict whether an alert notification is to be sent or not. An average mean of 95.8% accuracy was noted while using 60 decision trees in random forest algorithm. The average mean execution latency recorded for training and testing sets is 13.06 sec and 14.39 sec respectively. Observed accuracy rate, precision, recall and f1-score parameters were 95.8%, 96.5%, 98.5% and 97.2% respectively. Model buildup and the validation time computed were 3.26 sec and 4.25 sec respectively. It is also noted that at a threshold value of 0.93 in LS level, the maximum accuracy rate reached was 95.8%. Thus, based on the prediction of random forest approach, a decision to notify the users is taken. Obtained outcome indicates that the waste level can be efficiently determined, and the overflow of dustbins can be easily checked in time.

**Funding:** The author(s) received no specific funding for this work.

**Competing interests:** The authors have declared that no competing interests exist.

## 1. Introduction

A major concern for our environment in recent times is the rapid increase in garbage wastes, especially in urban regions. Regularly large volume of organic as well as inorganic wastes are generated at commercial and household places [1]. These waste products constitute degradable food materials, hospital related waste, household waste, plastics, bottles, and dead animals generated waste, industrial waste along with other commercial products. Dustbins are commonly used to gather this waste and when the volume of accumulated wastes rises then municipal units take charge to deal with it. At many places, the garbage containers are lying at public sites and household societies with an overflow of wastes. Lack of proper management of these waste lead to fatal health risk factors, which further may lead to spread of hazardous diseases. It also leads to pollute the overall environment [2]. As a result, the overall health and well-being of society is getting affected. Traditional approach of waste gathering and control is inefficient due to its poor waste collection management procedure. These methods have limited monitoring mechanisms available with poor waste gathering system and unsustainable throughput. Modern smart techniques are also deployed at few places but a problem faced by such smart waste control approaches is the rapid and continuous generation of massive data from sensor devices which is tough to track [3]. Similarly several other models using variety of technologies have been developed in the context of waste management. But most of these models lack proper synchronization regarding waste control and they are primarily applied for handling limited size homogeneous waste products. Moreover these models lack robustness to deal with heterogeneous garbage wastes. Few predictive models in recent times are designed but they fail to accurately categorize and predict distinct types of municipal wastes. Also, majority of existing hi-tech waste management modules are mainly utilized for data gathering and exchange within a restricted environment [4]. Automated models with least response time that can make proper use of the gathered data to make accurate and reliable informative prediction is yet to be deployed.

Advanced technologies, like Internet of Things (IoT) and artificial intelligence (AI) are quite powerful when they are applied individually [5]. Advancements in integrated hybrid technologies may be used to optimize the collection of these wastes, facilitating greater availability of service and thereby helpful in development of a smart and intelligent world in context to waste management systems. Of late, a cooperative merging of IoT with AI is explored, which is referred to as intelligence of things. In this integration, the IoT modules can be considered to be the digital neurons while AI models act as the system's brain. Here the requisite metrics are measured through sensory IoT units. These data are collected and processed by these IoT nodes, which are connected to distributed cloud storage. Later powerful AI based predictive machine learning algorithms can be used to analyze these stored data, develop relevant patterns from these stored data and automate the decision making process [6]. These systems are auto enabled to detect any abnormalities in data patterns thereby alerting users when deviation from usual trend is observed with minimal intervention of staff operators. These Intelligence of things enabled models can facilitate advanced level inter-connectivity among numerous services and applications in urban sustainable environment which range from industrial to societal issues [7]. Effective waste management is emerging out to be a chief factor as far as environmental sustainability is concerned. It is difficult to imagine a well-developed urban area without a smart waste management model in store. Advanced predictive capabilities of computationally intelligent algorithms can be integrated with data aggregation and monitoring by IoT sensor nodes to develop an accurate and reliable intelligent of things enabled model in effective waste management in sustainable smart city development.

In this paper, authors propose an automated and machine intelligent model based on intelligence of things to manage waste objects in a smart and sustainable environment. An

integrated framework of IoT and the random forest algorithm were used in a deployment of the hybrid intelligent model. Local sink nodes equipped with sensors and three smart bins each are deployed in sites to collect waste accumulated over time. With constant accumulation of waste in the smart bins, the threshold level is regularly observed. Sensors regularly monitor the values of smart bins and alert notifying messages are sent to concerned authority if the sensory data informs that the smart bins are filled up or any poisonous particles accumulates in the bins. Based on the smart bins values, LS weight and poison gas value recorded, the sink node is given more priority whose bins have crossed threshold level and need immediate attention to be emptied first. The random forest algorithm is used to classify, estimate, and predict the alert message decision that will be issued to the user.

The main contribution of the research is highlighted below.

♦ This research presents a new intelligence of things driven smart and predictive model to track and manage waste objects in a smart city environment. An integrated IoT driven predictive framework using random forest algorithm is used in deployment of the hybrid intelligent model.

♦ The model is comprised of static interconnected sensory bins attached to local sink node designed to determine waste level and load at any instant. When the maximum load limit is achieved, all waste related data is communicated to global sink node through internet connectivity.

♦ Aggregated data analysis and prediction is performed at global sink using random forest algorithm. Based on the recorded values of smart bins, sink weight and poison gas mass, priority is given to the sink node whose bins exceed threshold the level and needs to be immediately emptied.

♦ Variety of Sensors regularly monitor the values of smart bins and alert notifications are sent to concerned authority if the sensory data validates that the smart bins are filled up or any poisonous particles have accumulated in the bins.

♦ Thus, the smart and intelligent model can be efficiently used to classify waste, estimate time requirement for smart bins to be filled up, and notification of alert message decision to be issued to a user as well as official personnel, thereby automating waste regulation and restricting manual intervention.

The paper is laid out as follows. Section 1 highlights the intelligence of things concept and the need of effective waste management in smart environment. Section 2 presents the existing works related to the IoT based waste management systems in the Smart sustainable societies. Section 3 describes the proposed framework and its working features. Section 4 presents the implementation results with detailed analysis after deployment. Section 5 concludes the paper with the future work.

## 2. Literature survey

Various research works has been carried out with related to advanced waste management system in smart and sustainable environment. In [8], author has proposed an intelligent and automated dustbin monitoring model using decision tree classifier, which was an important step towards waste management. It was a smart waste monitoring model for wireless networks. Again in [9], the same author developed an integrated model, which was able to perform several tests runs on the previously built prototype by the usage of accelerometer, and magnetic proximity level. In [10] utilized an IoT based smart interface, which contained automated

space for virtual reality-based environment to simulate solid waste gathering. A RFID based waste monitoring framework was presented in [11] that took help of cell load sensing methodologies and digital assistants to achieve efficiency in waste control. Here RFID tags were embedded inside smart bins. Smart phones had a reader that transformed radio waves into digital data in the form of bin ID, which is reflected within digital assistants. In [12] highlighted a framework for multi agent simulation to integrate automated decision support to manage solid waste product in a distributed locality. Abstraction of geo-information system maps to 2D lattice was done to combine with multi agent model. An integrated model for waste status monitoring was developed in [13] where sensors were embedded in waste bins, which were further interconnected to cloud platform using push mechanism. All information was obtained from cloud. A garbage monitoring smart model was proposed in [14] in order to provide a sensing mechanism as to determine where the waste collected are wet or dry in nature and if the garbage waste lie outside the dustbins. In [15] framed an efficient waste gathering technique that designed an automated dustbin using IoT technology. Communication and data exchange through smart bin occurred by the use of ARM LPC 2148 using pressure sensing resistor and ultra-sonic sensors. An IoT enabled smart bin technique was developed in [16], which is useful in gaining free Wi-Fi access. Sensors present within the bin are used to detect the garbage level inside and whether they are filed up or not while free internet is provided to users through routers. In [17] proposed a dustbin interfaced with micro-controller based model with information retrieval wireless network depicting the present level of garbage on web browser triggered with HTML page of smart phones. In [18], authors have developed an adaptive feature optimization technique based on genetic algorithm for efficient classification of dengue patients. In [19], the authors have implemented some vital biologically inspired computation techniques to classify various types of tumours using Multi-layer perceptron as classifier. In [20] devised a GSM-based electronic monitoring system that sends an SMS to the supervisor when the dustbin is totally full, allowing the system to dispatch a trash collection truck. Once again, the supervisor is notified about the waste collection by SMS. In that work, the author employed an ultrasonic sensor to detect the level of trash(waste) in the dustbin, as well as a GSM module to provide information about the dustbin's state, such as whether it was full or empty. The author in [21] proposed a similar solution for waste collecting using an Arduino UNO board interfaced with a GSM module and an ultrasonic sensor, and also the author discussed the difficulties of smart dustbins such as its affordability, maintenance, and durability. When the waste are full upto the reference level, the ultrasonic sensor monitors the level of garage and guarantees that the dustbin is cleaned immediately [22,23]. In 2017, [24] presented an intelligent waste bin for the city of Pune in India, based on an IoT prototype. Authors in [25] proposed a waste level detection with collection scheduling of multiple bins. The system determines the location of three bins and estimates the waste amount in them using the Support Vector Machine (SVM) algorithm. It then used the Hidden Markov Model (HMM) to determine how many days are left before garbage needs to be collected. In [26] developed a secure authentication model for mobile cloud in context to key age, confirmation age and OTP generation. Here keys are developed using mobile identity number and SIM card ID. OTP uses client components and further possessed by PHP server. This model provides a superior security to customers that prevents from hacking. An application of human and robot interactivity in machine intelligence domain was proposed [27]. It presented a model for pest detection combining environment data with deep learning to facilitate farming community to examine trends in growth of crops to avoid early damage. A related study in [28] proposed a sensory based container to monitor and track food items present in home kitchen. It automated the food management process thereby tracking the level of foods available in the container but also generated a notification message when the items get expired. In [29], author

Table 1. Comparative analysis of existing works with their supporting features.

| | Bin Status | Waste Weight | Bin Location | Waste Level | Classify waste | Monitor |
|---|---|---|---|---|---|---|
| Force sensor and GSM/GPRS [30]. | Yes | No | Yes | Yes | Yes | Yes |
| RFID and Arduino Uno [31]. | Yes | Yes | Yes | Yes | Yes | Yes |
| GSM,Ultrasonic sensor and Arduino Uno, [32] | Yes | No | Yes | Yes | Yes | Yes |
| Ultrasonic sensor and WeMos [33]. | Yes | Yes | No | Yes | Yes | Yes |
| NodeMCU,Infrared sensors, air quality detector, (IoT)/GSM 5 [34] | Yes | Yes | Yes | Yes | Yes | Yes |
| Ultrasonic sensors, (IR) sensor and GSM module[35]. | Yes | Yes | Yes | Yes | Yes | Yes |
| LabView Tool, Arduino Mega and MQ-7 sensor [36] | No | Yes | No | Yes | Yes | Yes |
| servo motors, infrared radiation sensors, ultrasonic sensor, and (IR) sensors [37]. | Yes | Yes | Yes | Yes | No | Yes |
| GIS, GPRS, and RFID [38]. | Yes | No | Yes | Yes | Yes | No |
| Ultrasonic sensors, GSM, and Microcontroller [39]. | Yes | No | Yes | Yes | Yes | Yes |
| GSM kit, Arduino and Ultrasonic sensor [40]. | Yes | Yes | No | Yes | Yes | No |

has presented a multi-level IoT driven smart architecture for urban regions where effective management for wastes was demonstrated. The outcome showed that the model was scalable and can be applied to both indoor as well as outdoor settings with a less response delay.

As discussed above and summarized in Table 1, some existing models used different methodologies along with their distinguishing features. Some of these works use GSM and GPRS technology for waste management related data transfer. Slow data rate, limited bandwidth and less functional capacity are the main drawbacks of these models. Few other schemes are based on the usage of ultrasonic and infrared sensors. They suffer from very short testing range, error prone readings and are very inflexible. Also, they can be affected by surrounding conditions like rain and pollutants. Besides these limitations, not a single work is developed that makes use of predictive decision making of the aggregated data.

## 3. Materials and methods

At present waste are collected and removed in a manual way on regular basis. Special municipal personnel are assigned the job of monitoring and collecting waste at various regions on a daily basis. Some existing works have applied IoT based systems s to collect and monitor data from waste sites. Few machine learning algorithms had been used to predict and classify different kinds of waste products. But an integrated combination of both IoT and predictive analytics have seldom been deployed in real time. We have proposed an automated, reliable and priority based intelligence of things enabled waste management model with predictive capabilities in it.

### 3.1. Dataset used

The dataset for our research analysis is collected in real time environment from the sensory readings of the three smart bins and the sensors to detect hazardous gas elements presence. A total of 510 instances were aggregated altogether. The waste collected in smart bins are parameterized with two variables, which include bin's height and bin's mass. The values are categorized into normal, moderate and peak labels as summarized in Table 2.

Based on the categorized labels and sensory values aggregated, a total of 9 distinct scenarios are feasible, which is shown in Table 3.

The gas sensors used in the study to detect poisonous elements like nitrogen dioxide, carbon monoxide and methane in garbage waste are MQ-2, MQ-136, and MICS-2174

**Table 2. Defining threshold range for smart bins parameters.**

|  | Normal | Moderate | Peak |
|---|---|---|---|
| **Smart Bin Height (cm)** | 40–60 cm | 20–40 cm | 1–20 cm |
| **Smart Bin Mass (kg)** | 1–4 kg | 4–10 kg | 10–18 |

**Table 3. Scenarios to bin metrics mapping and labeling.**

| Scenario | Height (H) | Mass (M) | Garbage Level (GL) |
|---|---|---|---|
| Scenario 1 | Normal | Normal | 0 |
| Scenario 2 | Normal | Moderate | 0 |
| Scenario 3 | Normal | Peak | 1 |
| Scenario 4 | Moderate | Normal | 0 |
| Scenario 5 | Moderate | Moderate | 1 |
| Scenario 6 | Moderate | Peak | 2 |
| Scenario 7 | Peak | Normal | 1 |
| Scenario 8 | Peak | Moderate | 2 |
| Scenario 9 | Peak | Peak | 2 |

**Table 4. Poisonous gas range and threshold details.**

|  | NO2 | CO | CH4 |
|---|---|---|---|
| **Concentration level (ppm)** | 0.25–5 | 20–1000 | 300–10000 |
| **Predefined Threshold level (ppm)** | 2.625 | 510 | 5150 |

respectively. The degree of concentration for all these hazardous gases along with their predefined threshold level are shown in Table 4.

Table 5 depicts a sample scan demonstration readings of the waste dataset collected in real time environment. The data values are gathered from different sensors corresponding to the parameters of three smart bins, local sink (LS) and the poisonous gas level. The smart bins comprises of height of bin and mass of bin as its two parameters.

## 3.2. Proposed Intelligence of Things based Waste management model

The skeleton framework of the intelligent and smart model is illustrated in Fig 1. The model consists of multiple Local Sink (LS) and a single Global Sink (GS). Each LS is connected with 3 smart bins. A smart bin is allocated its corresponding LS based on the least distance criteria. Every smart bin is assigned a unique ID. The smart bin is static and fixed to a certain area. Details regarding the smart dustbins and their location are maintained in a database. The dustbin is equipped with load sensors and ultraviolet sensors to detect the level of waste accumulated in the dustbins. It has humidity sensor for the garbage detection. All dustbins communicate with Raspberry-pi 3, which acts as a broker. The task of Raspberry-pi is to gather all sensor specific data from the smart dustbins and transmit this data to a server with the use of Wi-Fi. The details comprises of the unique ID of dustbins, garbage level, and humidity content in the waste. The IDs are matched with dustbin database in the server thereby determining the dustbins levels present in various regions.

Smart bins are equipped with sensors to capture weight of the bin at regular intervals and their location can be determined at any instant. The role of LS is to collect and aggregate the

**Table 5. Garbage waste dataset sample collected from sensors for the proposed model.**

| Smart Bin 1 | | | Smart Bin 2 | | | Smart Bin 3 | | | LS | Poisonous Gas Level | | |
|---|---|---|---|---|---|---|---|---|---|---|---|---|
| H(cm) | M(kg) | GL | H(cm) | M(kg) | GL | H(cm) | M(kg) | GL | M(kg) | NO2 (ppm) | CO (ppm) | CH4 (ppm) |
| 16.5 | 15.8 | 2 | 43.3 | 3.5 | 0 | 41.3 | 3.2 | 0 | 3.7 | 0.37 | 84 | 1000 |
| 48.7 | 15.5 | 1 | 46.5 | 3.2 | 0 | 45.5 | 3.1 | 0 | 4.8 | 0.46 | 200 | 2000 |
| 33.4 | 11.2 | 2 | 42.5 | 14.1 | 1 | 42.8 | 14.6 | 1 | 6.4 | 0.52 | 100 | 1500 |
| 52.1 | 1.9 | 0 | 14.7 | 11.4 | 2 | 32.8 | 5.7 | 0 | 8.3 | 0.29 | 300 | 1800 |
| 12.7 | 5.9 | 2 | 43.4 | 11.7 | 1 | 24.4 | 3.7 | 0 | 9.2 | 1.3 | 200 | 2100 |
| 17.5 | 7.5 | 2 | 34.5 | 3.5 | 0 | 12.7 | 13.2 | 2 | 9.6 | 1.6 | 220 | 2400 |
| 15.7 | 7.6 | 2 | 26.3 | 7.4 | 0 | 32.1 | 3.8 | 0 | 10.2 | 2.2 | 150 | 3000 |
| 27.9 | 13.8 | 2 | 29.4 | 16.8 | 2 | 32.6 | 8.4 | 1 | 10.3 | 0.86 | 310 | 3100 |
| 41.3 | 3.6 | 0 | 42.3 | 13.2 | 1 | 26.8 | 17.9 | 2 | 10.5 | 2.6 | 420 | 2600 |
| 41.8 | 3.7 | 0 | 52.4 | 15.3 | 1 | 52.1 | 15.9 | 1 | 10.8 | 3.1 | 300 | 2800 |
| 43.5 | 2.1 | 0 | 18.2 | 15.4 | 2 | 18.5 | 13.4 | 2 | 11.3 | 2.7 | 260 | 3100 |
| 32.8 | 5.7 | 0 | 13.4 | 6.8 | 1 | 13.9 | 4.8 | 1 | 11.6 | 1.8 | 170 | 3000 |
| 24.4 | 3.7 | 0 | 42.8 | 14.2 | 1 | 35.8 | 16.5 | 2 | 11.9 | 3.3 | 250 | 2000 |
| 12.7 | 13.2 | 2 | 34.4 | 3.7 | 0 | 18.5 | 17.8 | 2 | 12.2 | 2.9 | 360 | 1000 |
| 32.1 | 3.8 | 0 | 32.1 | 3.8 | 0 | 16.8 | 7.3 | 2 | 12.4 | 2.4 | 100 | 1500 |
| 32.6 | 8.4 | 1 | 32.6 | 8.4 | 1 | 25.4 | 12.8 | 2 | 12.6 | 3.2 | 200 | 1900 |
| 26.8 | 17.9 | 2 | 26.8 | 17.9 | 2 | 26.8 | 17.9 | 2 | 12.9 | 1.4 | 100 | 2200 |
| 45.2 | 14.2 | 1 | 45.2 | 14.2 | 1 | 45.2 | 14.2 | 1 | 13.2 | 0.9 | 90 | 2800 |
| 41.4 | 11.7 | 1 | 41.4 | 11.7 | 1 | 41.4 | 11.7 | 1 | 14.5 | 2.5 | 220 | 2900 |
| 33.5 | 3.9 | 0 | 33.5 | 3.9 | 0 | 33.5 | 3.9 | 0 | 14.6 | 1.7 | 400 | 3000 |
| 26.6 | 7.4 | 0 | 26.6 | 7.4 | 0 | 26.6 | 7.4 | 0 | 14.7 | 3.6 | 200 | 1300 |
| 29.1 | 16.8 | 2 | 29.1 | 16.8 | 2 | 29.1 | 16.8 | 2 | 15.6 | 4.2 | 320 | 1700 |
| 42.7 | 13.2 | 1 | 42.7 | 13.2 | 1 | 42.7 | 13.2 | 1 | 16.4 | 1.7 | 380 | 1900 |
| 52.1 | 15.9 | 1 | 52.1 | 15.9 | 1 | 52.1 | 15.9 | 1 | 17.3 | 3.8 | 390 | 2000 |
| 18.5 | 15.4 | 2 | 18.5 | 15.4 | 2 | 18.5 | 15.4 | 2 | 17.5 | 0.75 | 280 | 2300 |
| 13.9 | 6.8 | 1 | 13.9 | 6.8 | 1 | 13.9 | 6.8 | 1 | 17.7 | 0.92 | 190 | 2400 |

data received from different smart bins associated with it. Different types of waste are regularly deposited within these smart bins. When a bin has received its maximum limit of waste, its sensor is activated and it signals this data to its corresponding LS through a weight parameter. The LS records this weight value in its database and allocates a unique timestamp (TS) to that smart bin at that instant. Similarly, unique TS is allocated to all smart bins associated with LS once they are completely filled up with waste. The TS values of all smart bins are gathered at their respective LS. Then this data is transmitted to the GS through internet connection. Detailed data analysis is done at GS. This analysis involves calculation of TS values with respect to time and cumulative wt-max index of individual LS. With efficient data analysis, it can be predicted that the time interval of each smart bin to be completely filled up with waste. When this information is received by the GS, the vehicle rushes to the location and removes all waste from the dustbin. When it is known how fast a smart bin is getting filled, it can be either replaced or additional smart bins can be deployed at that area. Thus, this model helps to automate the waste management process thereby reducing the manual effort by a significant level.

The working flow of the developed model with its distinguishing steps are presented in Fig 2. The subsequent vital modules are described here.

**LS proximity based automated lid opening.** An automated and smart waste management model is developed. The lid of Local sink (LS) is made automatic. The LS has a front-

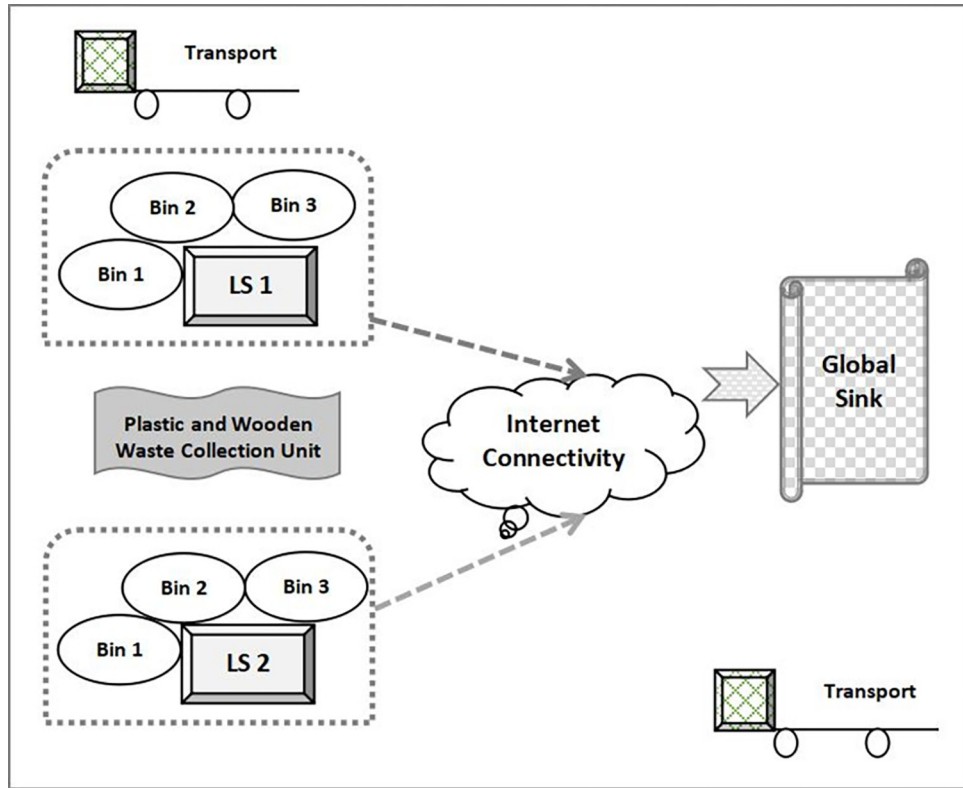

**Fig 1. Architecture of automatic waste management system.**

facing IR sensor that detects the user's proximity and automatically opens and closes the lid. The linear motion required to open and close the lids of the smart dust bin is controlled by actuators.

**Determine LS current status.** The current status of LS is determined so that accordingly the user is able to figure out the quantity of waste that can be further deposited. Status of LS includes the current weight of LS and presence of different types of waste or poisonous elements in smart dustbins. To accomplish this task, a blue button is available and when the user clicks that button all kinds of updated information regarding the dustbin and LS come up.

**Waste deposition and accumulation.** The IR sensor, as well as the three ultrasonic sensors inside the three distinct bins, constantly monitors the level of garbage and waste in the dustbin compartment. If the LS is not found to be full then user can push waste and garbage onto the LS. Garbage is placed onto the LS's conveyer belt, and the presence of waste is first detected using an infrared sensor at the conveyor belt's start end.

This way the garbage gets gradually deposited and accumulated.

**Waste detection and segregation.** The inductive proximity sensor detects whether the waste that moves further is metal. If metal is detected, the electromagnet changes its direction for metallic waste collection. After that, the garbage is demagnetized and placed in bin 1. The conveyer is then moved forward, and the dry waste is blown out by an air blower. Light particles such as plastic, paper, and other materials are separated and deposited in bin 2. The moisture sensor detects the wet garbage, and it is dropped into bin 3 when the conveyor belt goes forward. The detection of plastic and wooden waste is handled by the capacitance proximity sensor on the conveyer belt. It is moved to the plastic and wooden collecting units to empty

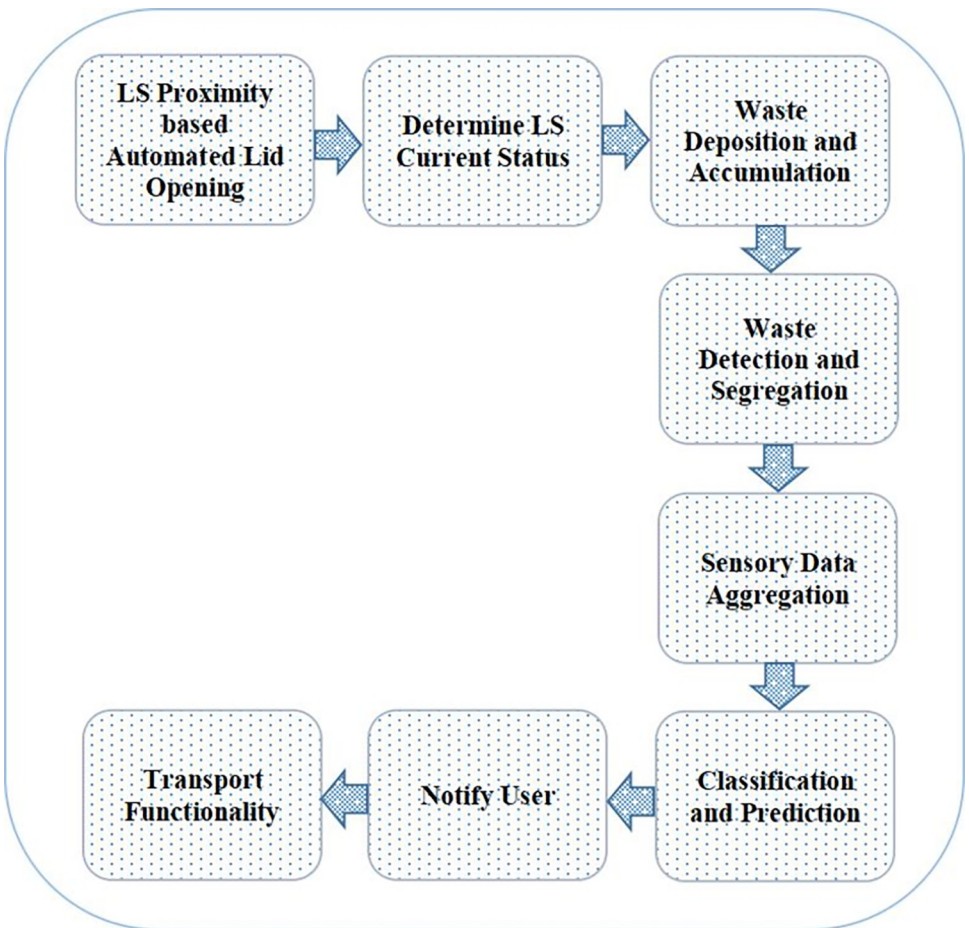

**Fig 2. Work flow modules of the developed model.**

them as soon as it detects plastic or wood. A gas sensor is used to identify harmful gases in garbage, such as methane.

**Sensory data aggregation.**    After implementing different functionalities associated with the smart waste management, further the sensor values are to be collected. The gathered sensor values include bin 1 level, bin 2 level, bin 3 level, LS weight and poison gas value. These sensory data values are collected together in a comma separated file (csv) format by LS. The accumulated csv file is sent is pushed to GS for analytical processing [41–43].

**Classification and prediction.**    The GS is responsible for processing the aggregated sensory data received from LS in csv format. According to Eq 1, these sensory data values are normalised from 0 to 1.

$$N(F_i) = \frac{F_a}{F_{max}} \tag{1}$$

Where $F_a$ is the original value acquired by the sensor, and $F_{max}$ is the parameter's maximum predicted value. Different service providers can set the maximum predicted value. The decision to send or not send a weather alert message is made automatically based on combinations of N (Si). The entire dataset is divided into training and testing samples in 60:40 ratio. The decision to send or not send a weather alert message is made automatically based on a combination of sensor data values. Random forest algorithm is used to predict decision directly from

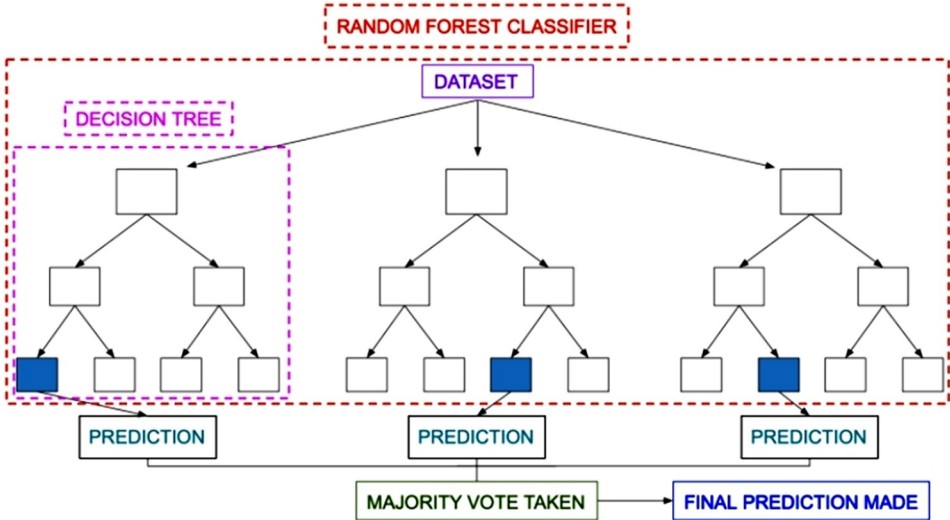

**Fig 3. Random forest model demonstration.**

the training data set. Then it is used for classification, estimation and prediction of alert message decision [44–47].

The ensemble-based classifier used in this study constitutes the integration of separate decision trees so as to form a random forest [48]. Here individual tree generates a label prediction and the class with the most votes become the final predicted label of the model. Each separate tree denotes an unrelated model with other trees in the forest. As a pool of multiple uncorrelated models random forest outperforms other individual decision tree-based models. The ensembled predicted outcome generated by uncorrelated models provides more correct values than other individual predictions as a result of randomness of variables. A simple representation of random forest is shown in Fig 3.

The steps involved in the working process of random forest are explained below [49,50].

1. Take K data points at random from the training set.

2. Create decision trees for the K chosen data points (Subsets).

3. Choose N for the number of decision trees to be created.

4. Repeat Step 1 & 2.

5. Find the predictions of each decision tree for new data points, and allocate the new data points to the category with the most (wins majority) votes.

   For random forest methods, the pseudocode can be divided into two parts.

- Pseudocode to create the random forest by merging N decision trees).

- Pseudocode to predict using the created random forest classifier bymaking predictions for every tree in the random forest).

```
Pseudocode for Random Forest creation:
1. Select "k" features at random from a total of "m" features, where k
<< m.
2. Calculate the node "d" (root node) using the optimal split point
among the "k" features.
```

```
3. Using the best split, divide the node into daughter nodes (sub
nodes or child nodes).
4. Repeat steps 1-3 until the "l" number of nodes is attained.
5. Repeat steps 1-4 for "n" number of times to create a random forest
with "n" number of trees.
```
**Pseudocode for prediction:**
```
1. Consider the test features.
2. Use the rules of the individual random generated decision tree to
predict result.
3. Then save the predicted outcome.
4. Compute the number of votes for each individual predicted outcome.
5. Consider the random forest algorithm's final prediction to be the
predicted outcome that has majority of votes.
```

It is very important in the random forest classifier to identify a feature. Various methods like information gain (IG), and Gini index (GI) can be computed for getting the most informative feature (feature that yields the most information (IG). Once the scores for all of the available features have been calculated, the model will choose the feature with the highest score for each root node.

The information gain (IG), and Gini index (GI) are shown in Eqs 2 and 3 respectively.

$$IG\left(D_p, f\right) = I\left(D_p\right) - \sum_{j=1}^{m} \frac{N_j}{N_p} I(D_j) \tag{2}$$

Where,

$f$ = the feature to perform the split,

$D_p$ = data set of the parent,

$D_j$ = $j$-th child node,

$I$ = the impurity measure,

$N_p$ = the total number of samples at the parent node, and

$N_j$ = the number of samples in the $j$-th child node.

$$I_{GINI} = 1 - \sum_{i=1}^{j} p_i^2 \tag{3}$$

Where,

$j$ = the number of classes present in the node.

$p$ = the distribution of the class in the node.

**Notify users.** This functionality handles the notification of alert message to be sent to users. Notification is sent to users if the dustbin is filled up or any poisonous elements are detected, or the weight of dustbin exceeds the threshold value. Based on the prediction of random forest approach, decision is taken as to notify the user or not.

**Transport functionality.** Once it is determined that the dustbin in LS is filled up or any other casualty has happened then GS activates the transport function, and the transport vehicle turns up soon to empty the smart bins and resolve any other issues.

## 4. Results and discussion

The proposed intelligence of things enabled smart waste management model was implemented in a local society in Bhubaneswar, Odisha. It was able to prevent overflow in dustbins and transmit alert message when the dustbins are full. As soon as the dustbins near the threshold limit, it gets connected to the network and an alert message is sent to the concerned base station where the user monitors the status. It helps in real-time transmission and avoids unnecessary delay.

## 4.1. Smart bins load analysis

A sample case study analysis was carried out in a locality to test its effectiveness. A total of 15 smart bins were scattered in the locality to collect wastes from its respective region for a time period of 6 hours at a stretch. All the bins were allotted a unique ID (B1, B2, B3. . ...B15). The dustbin load status is calculated regularly at an interval of 2 hours. The threshold value for bin full load is reserved at 20 waste units.

As it can be seen from Fig 4, after 2 hours interval B2, B10 and B14 are completely filled up while B9 and B13 are nearing to be filled up. It signifies the rate of wastes deposited in these filled up bins are very high due to its local population and waste deposition strength. B6, B7, B12 and B15 are least filled bins due to their lower frequency of usage.

Similarly in Fig 5 it is observed that by the completion of 4 hours of deployment B3, B9 and B13 have reached the threshold limit value and can no more take further solid wastes. The global sink is instantly notified whenever any bin gets filled up or unable to take more waste. The variation is time for the bins to get filled up is due to factors like population capacity of the locality, types of waste accumulated in bin or distance of the bin from a locality.

At the end of full 6 hours it is seen that B1 and B11 have reached the threshold value and the bins are full as seen in Fig 6. Since the nearby bins are already filled up so the wastes are accumulated in farther placed bins.

Thus from the bin load analysis, it is observed that bins which are near to the society (B1, B2, B3,9, B10, B11, B13 and B14) are used more frequently and thus gets filled up quickly as compared to those present at farther areas from society (B4, B5, B6, B7, B12 and B15). Also among the nearby bins, few bins like B2, B10 and B14 took heavy load initially as a result got filled up in less than 2 hours time. Hence it is suggested to place the bins as close as possible so that the society people can utilize those bins conveniently. Also the size of few bins should be increased so as to be able to take more load at quick intervals.

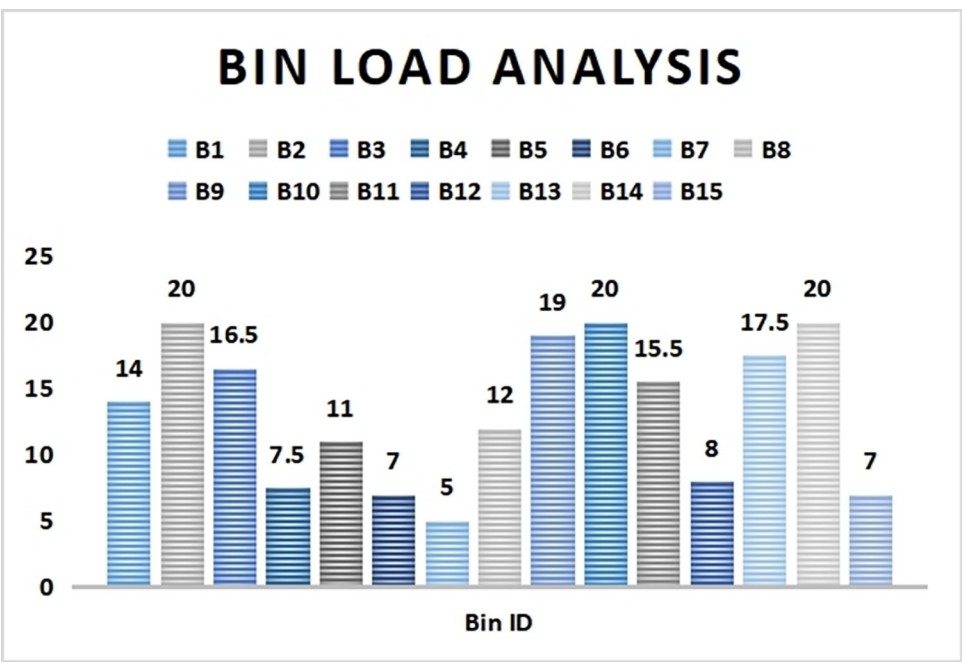

**Fig 4. Bin load status after 2 hours of deployment.**

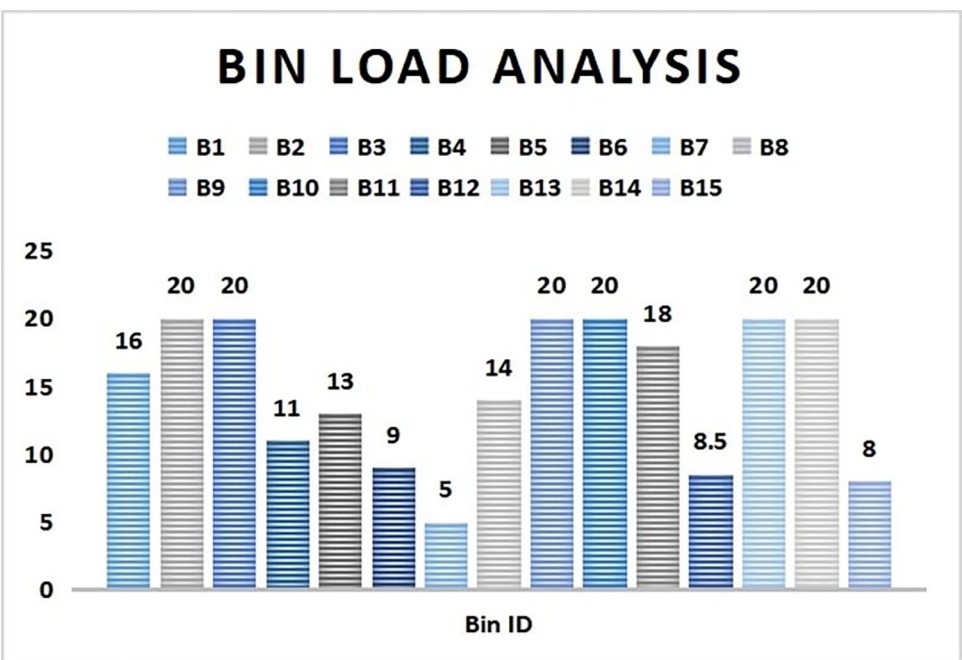

**Fig 5. Bin load status after 4 hours of deployment.**

A latency period analysis is undertaken to determine the time interval taken by the smart bins to be filled up. All the bins are placed at close proximity of the society on a very busy holiday. It is seen from Fig 7, B11 takes the least time period of 15 min to reach the threshold value of 20 waste units and B14 takes the maximum time of 153 min to be filled up with solid wastes.

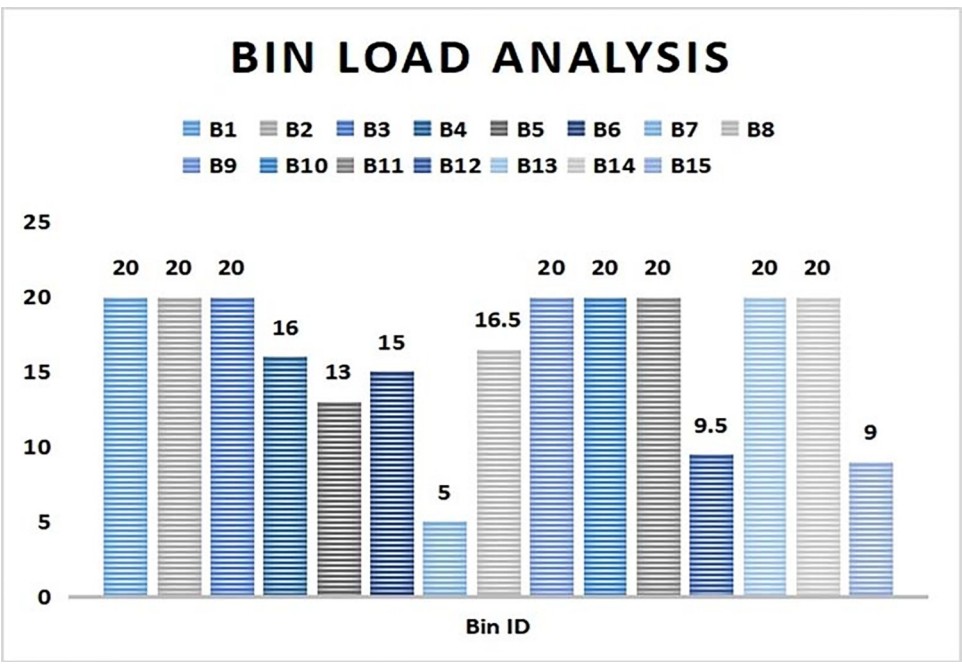

**Fig 6. Bin load status after 6 hours of deployment.**

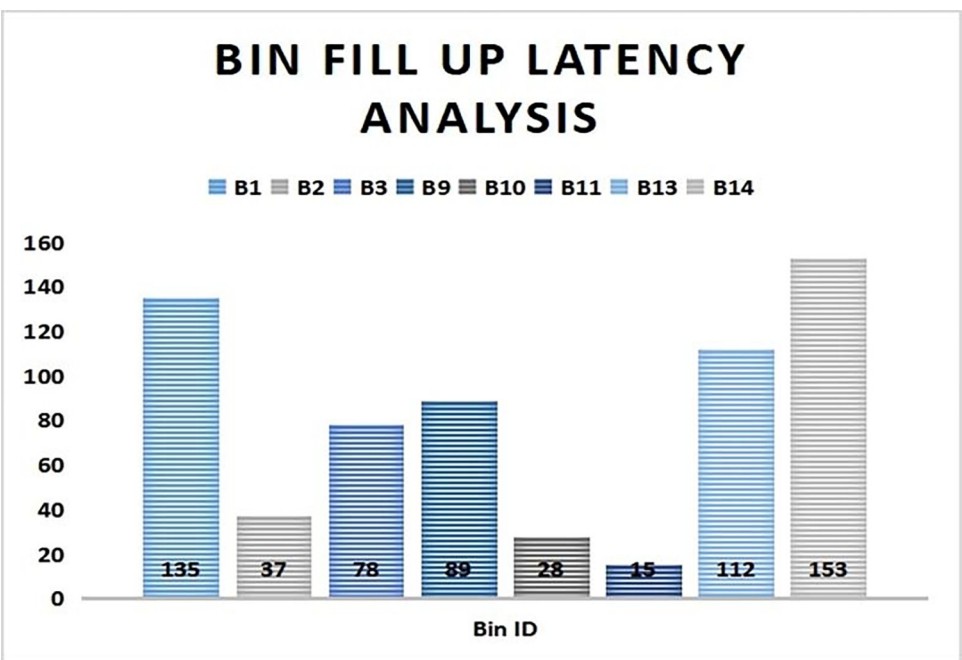

**Fig 7. Time analysis of bin fill up status.**

Hence it can be inferred that due to the less volume structure of the bin B11, waste consumption by B11 is much more compared to others. Since B14 is a larger bin so it takes longer time to be filled up. Thus it is wise to position more larger sized bins in the same locality.

Table 6 shows the bin load status of smart dustbins that never reached the threshold value and were not filled up after 6 hours time. As it is seen, bin B7 got hardly any waste into it over a length of time due to its farther distance location and low waste accumulation frequency. Its waste unit was constant with 5 units which shows that the locality was very sparse with very less generation of solid waste in that region.

## 4.2. Predictive performance analysis using random forest

This sub section illustrates the predictive analysis of the intelligent of things based waste management model using random forest method. Various evaluation metrics are considered for demonstrating the model's performance [51].

A classification accuracy analysis was performed using the proposed smart model with respect to varying the number of decision trees used as shown in Fig 8. A maximum accuracy rate of 97.9% was recorded when as many as 20 decision trees were implemented while a relatively lower accuracy of 93.8% was obtained with 5 number of decision trees used. Thus, a simple average mean value of 95.8% was noted and a total number of 60 decision trees were used in random forest algorithm.

**Table 6. Bins that were never filled up after 6 hours.**

| Bin ID | B7 | B12 | B15 |
|---|---|---|---|
| Load after 2 hours | 5 | 8 | 7 |
| Load after 2 hours | 5 | 8.5 | 8 |
| Load after 6 hours | 5 | 9.5 | 9 |

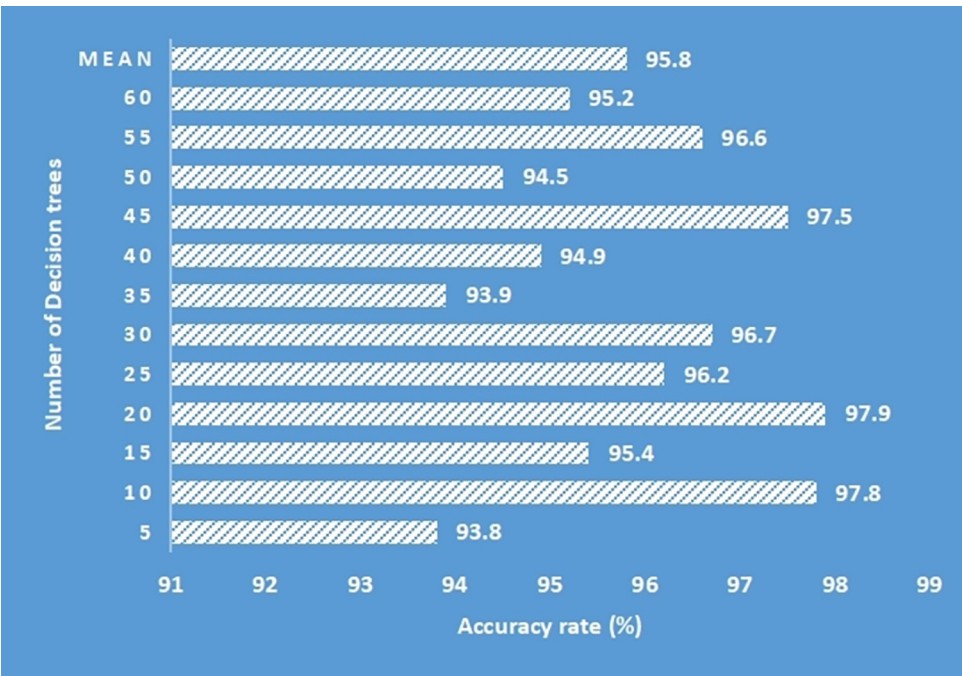

**Fig 8. Classification accuracy analysis with respect to decision trees used in random forest algorithm.**

An execution time analysis was undertaken as far as training and testing latency period are concerned. Again, the range of decision trees taken into consideration in random forest algorithm is from 5 to 60. though the execution time increases with the increase in number of decision trees still the rise is manageable and is linear in nature. Training and testing time during initial stage was only 2.2 sec and 2.4 sec respectively. Towards the end of implementation when the decision trees count reached upto 60, the training and testing time were noted to be 26.2 sec and 27.92 sec respectively. The average mean execution latency observed for training as well a testing set is 13.06 sec and 14.39 sec respectively. The outcome is shown in Fig 9.

Several performance metrics, like accuracy, precision, recall and f1-score were used for the validation of the proposed waste management model [52]. Different machine learning algorithms, like neural network, regression, KNN, SVM, and decision tree, were used for comparison purpose. SVM recorded an overall inferior performance as compared to other algorithms. The proposed smart model outperformed other algorithms and generated an optimum value. The accuracy rate, precision, recall and f1-score parameters were 95.8%, 96.5%, 98.5% and 97.2% respectively as highlighted in Table 7.

The model build-up time and validation time comparative analysis for the proposed model was carried out with other algorithms and is depicted in Fig 10. Neural network gave the maximum time period for predicting the outcome while the proposed model generated the least time to take decision to send the alert notification or not. The build up and the validation time recorded using the proposed model was 3.26 sec and 4.25 sec respectively. Since no feature scaling is needed, thus it is quite fast.

Reason for superior performance of a random forest model is due to many factors. It can easily handle high dimensional dataset as it can work with multiple subsets of data. Random forest smoothly deals with outliers by binning them. It can also be effective in balancing errors with unbalanced data samples. It can also automatically handle missing values. It has a low bias compared to others generating higher accuracy rate. The overall performance of the

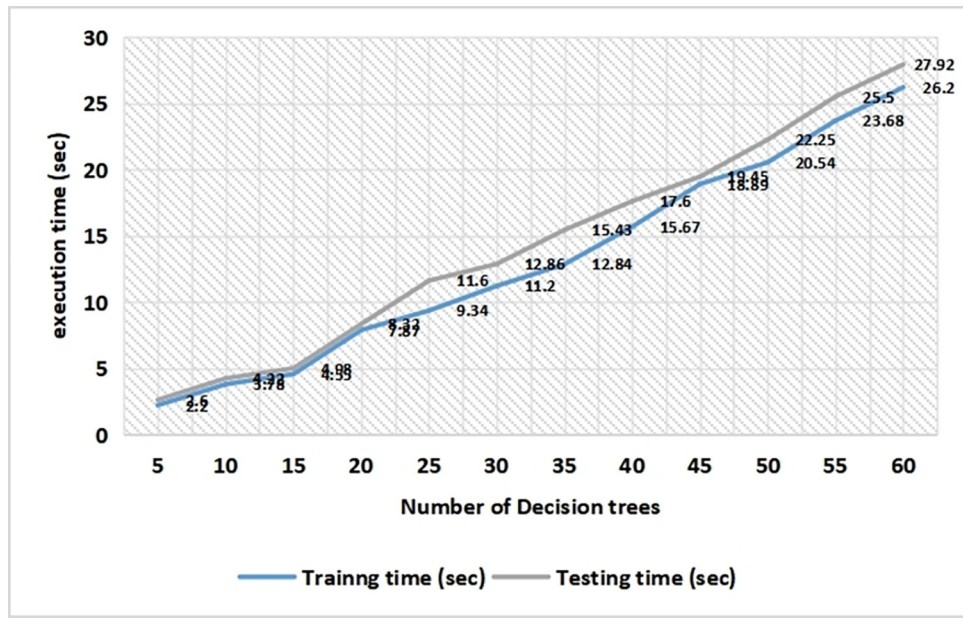

**Fig 9. Execution delay analysis with respect to decision trees used in random forest algorithm.**

**Table 7. Performance metrics comparison with different predictive models.**

|  | Accuracy | Precision | Recall | F1-Score |
|---|---|---|---|---|
| Neural Network | 94.2% | 93.2% | 94.6% | 94.2% |
| KNN | 95.1% | 95.4% | 96.5% | 96.2% |
| Regression | 92.8% | 92.6% | 93.8% | 93.1% |
| Support Vector Machine | 88.2% | 86.2% | 90.2% | 88.8% |
| Decision tree | 92.5% | 91.7% | 94.6% | 93.1% |
| Proposed Random forest based Model | 95.8% | 96.5% | 98.5% | 97.2% |

model was also dependant on the threshold value of the local sink. It is the upper bound value considered for the local sink to be filled up before notification is sent to users. It is noted that at a threshold value of 0.93, the maximum accuracy rate reached was 95.8% while the accuracy decreases with a steady decrease in threshold value. The least accuracy of 72.5% was observed with a 0.69 threshold value of local sink. The result is shown in Fig 11.

## 4.3. Benefits of the proposed intelligence of things based waste management model

➢ The developed model is more effective and time saving in automation of solid waste coordination and managing overall aggregation process thereby restricting manual intervention to a large extent.

➢ The smart bins and garbage level in the bins can be continuously monitored at global sink in an uninterrupted manner facilitating a more sustainable throughput.

➢ The model is made to be fault tolerant since accidental failure of few bins will not affect the overall waste management process.

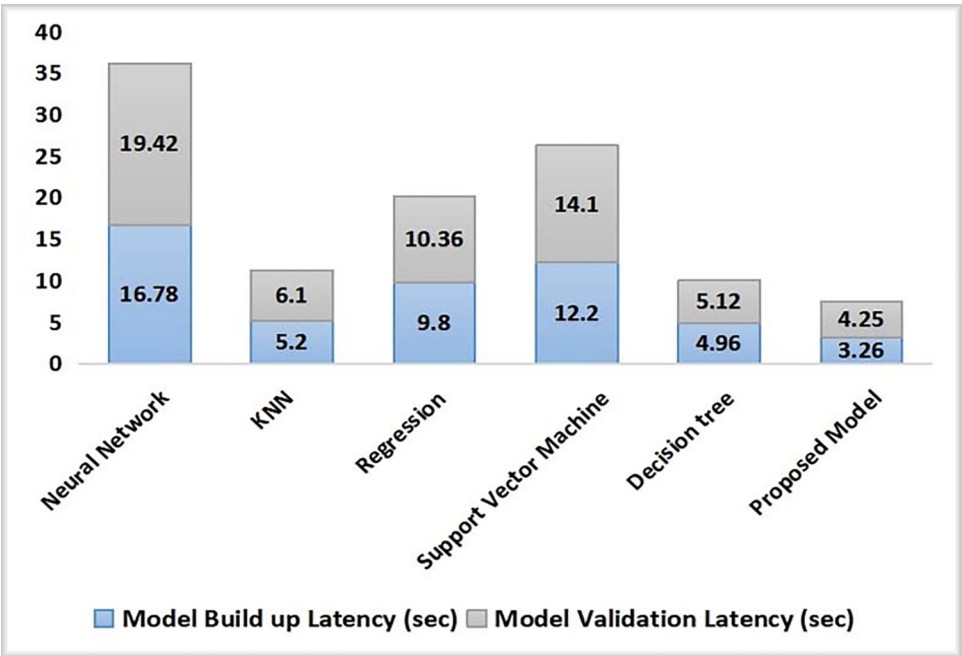

**Fig 10. Model training and testing latency analysis with different algorithms.**

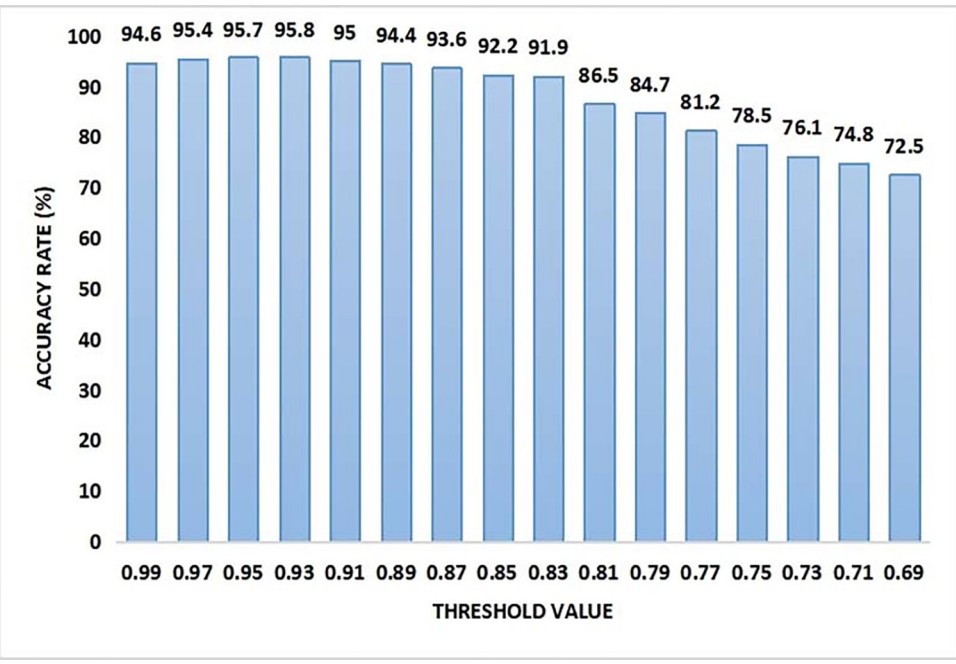

**Fig 11. Accuracy analysis in context with the threshold value of smart bin.**

➤ An acknowledgement link is included in the model where the local sink is acknowledged when it gets filled with wastes and this is communicated to the global sink.

➤ The model provides superior synchronization and awareness regarding waste control and it can handle a wide range of waste products with its distributed cloud storage.

➤ Population density can be clustered and segregated so that dense populated regions can be identified and may be provided with more bins of increased volume to accept more load.

➤ The developed model can not only be utilized for data gathering and exchange within a restricted environment but also it acts as a decision making module to classify the wastes as well as predict the accumulation of wastes and smart bins requirement in a concerned region.

➤ An alert unit is also associated with the model which notifies the user in case of any discrepancy, the bins are completely full or any poisonous particles presence.

➤ The overall waste management system becomes more reliable, robust, accurate and productive.

## 5. Conclusion

Waste management is a critical domain in urban areas where sustainability is a factor. In our research, an integrated hybrid model using intelligence of things and machine learning for smart automated waste monitoring and management system is designed and implemented. The smart bins collect bin load status data and send them to the corresponding local sink node. The local sink nodes aggregate the collected information and transmits it to the global sink station for further processing and analysis. Based on the timestamp of the smart bins, the priority of the bins was computed. The model was implemented in locality in Bhubaneswar, Odisha, India. The result obtained was very constructive. Smart bins closer to densely populated sites were filled up at a faster rate compared to that of sparse population. The random forest algorithm was used as a classification method to determine whether an alert message is to be communicated to users or not. A mean classification accuracy of 95.8% was observed with 60 decision trees of the random forest algorithm. The mean latency delay time noted for training and testing data is 13.06 sec and 14.39 sec respectively. The accuracy rate, precision, recall and f1-score metrics recorded were 95.8%, 96.5%, 98.5% and 97.2% respectively. The model training time and validation time recorded using the proposed model was 3.26 sec and 4.25 sec respectively, which is quite less when compared with other existing approaches. It is also noted that at a threshold value of 0.93 in a local sink level, the maximum accuracy rate reached was 95.8%.

Thus, based on the prediction of random forest approach, a decision can be taken as to notify the user or not. The presented model was completely automated with the least amount of human interaction. Overall, the proposed system to detect and manage waste was proven to be very effective and intelligent. The proposed system functions for gathering and updating waste related data automatically as well as processes the utilization of data in an intelligent manner. Internet usage here enhances the efficiency and reliability of the system with long distance coverage. Hence, it may be useful for government agencies for efficient monitoring of solid waste and management in urban populated areas, near medical centers or educational sites where solid waste generation is quite high.

This system can be further upgraded to be implemented in large scale with further inclusion of more parameters and more security features along with fault tolerant protocols can be

integrated to make it more efficient. Also it can be enhanced to accept image based datasets for waste assessment. Besides, the model can be mapped onto a mobile application where every data will be at finger tip of users.

## Supporting information

**S1 File.**
(ZIP)

## Author Contributions

**Conceptualization:** Sushruta Mishra, Hrudaya Kumar Tripathy.

**Data curation:** Sushruta Mishra, Lambodar Jena.

**Investigation:** Sushruta Mishra, Lambodar Jena.

**Methodology:** Sushruta Mishra, Lambodar Jena, Hrudaya Kumar Tripathy.

**Project administration:** Hrudaya Kumar Tripathy.

**Resources:** Tarek Gaber.

**Software:** Sushruta Mishra, Lambodar Jena.

**Supervision:** Hrudaya Kumar Tripathy, Tarek Gaber.

**Validation:** Hrudaya Kumar Tripathy, Tarek Gaber.

**Writing – original draft:** Sushruta Mishra.

**Writing – review & editing:** Hrudaya Kumar Tripathy, Tarek Gaber.

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
