## [Decision Letter · Decision Letter 0]

9 May 2022

PONE-D-22-03543Prioritized and Predictive Intelligence of Things Enabled Waste Management Model in Smart and Sustainable EnvironmentPLOS ONE

Dear Dr. Gaber,

Thank you for submitting your manuscript to PLOS ONE. After careful consideration, we feel that it has merit but does not fully meet PLOS ONE’s publication criteria as it currently stands. Therefore, we invite you to submit a revised version of the manuscript that addresses the points raised during the review process.

ACADEMIC EDITOR: Please revise the paper according to the comments provided by reviewers and make each explicitly. 

We look forward to receiving your revised manuscript.

Kind regards,

Chakchai So-In, Ph.D.

Academic Editor

PLOS ONE

Journal Requirements:

Reviewers' comments:

Reviewer's Responses to Questions

**Comments to the Author**

1. Is the manuscript technically sound, and do the data support the conclusions?

Reviewer #1: Yes

Reviewer #2: Yes

2. Has the statistical analysis been performed appropriately and rigorously? 

Reviewer #1: Yes

Reviewer #2: Yes

3. Have the authors made all data underlying the findings in their manuscript fully available?

Reviewer #1: Yes

Reviewer #2: Yes

4. Is the manuscript presented in an intelligible fashion and written in standard English?

Reviewer #1: Yes

Reviewer #2: Yes

5. Review Comments to the Author

Reviewer #1: The paper is in general well-written, has merit and I think can be published, after a minor review. Some minor suggestions for enhancement are:

1 – The authors could summarize the introduction, it is too long in the present form.

2 – I suggest the authors include the following paper in their literature review: “An IoT-based smart cities infrastructure architecture applied to a waste management scenario”, Ad Hoc Networks, Volume 87, 2019, Pages 200-208, ISSN 1570-8705, https://doi.org/10.1016/j.adhoc.2018.12.009.

3 – Section 3 could be better structured, divided in subsection according to the subjects handled in the text.

4 – The results are well presented but not so nicely discussed in Section 4. I recommend the authors to elaborate better on the discussion.

5 – I suggest the authors divide the text in the conclusion in at least 3 paragraphs. Two for the conclusion, and a third, and last one, for discussions on future works.

Reviewer #2: *Numerous solutions to this problem have been proposed in past research. Please explain novelty and contribution of the study. Why this study is important and what academic values it adds to the field?

*The introduction needs to be rewritten more precise and concrete and providing the much better motivation, significance and impact of the paper. Please explain research gap explicitly then propose research questions.

* Include the below references to support your claim

S.Shanmuga Priya, A.Valarmathi, M.Rizwana, L.Mary Gladence, “Enhanced Mutual Authentication System in Mobile Cloud Environments” in International Journal of Engineering and Technology ISSN: 2227-524X, Vol.7 No.3.34 2018,Page.No 192-197

Mary Gladence, L., Vakula, C.K., Selvan, M.P., Samhita, T.Y.S., “A research on application of human-robot interaction using artifical intelligence” in International Journal of Innovative Technology and Exploring Engineering (IJITEE) ISSN: 2278- 3075, Volume-8, Issue- 9S2, July 2019.

Jinila, Y. Bevish, V. Rajalakshmi, L. Mary Gladence, and V. Maria Anu. "Food Consumption Monitoring and Tracking in Household Using Smart Container." In Proceedings of the Third International Conference on Computational Intelligence and Informatics, pp. 693-700. Springer, Singapore, 2020

*.The discussion of Experiment should be written more in-depth, more precise and concrete, such as what questions were resolved? How can the proposed method solve these problems? Then, the advantages (and disadvantages?) of the proposed methods should be discussed.

*In the conclusion, the practical application field of the proposed methods and the research findings can be described that highlights the contribution of this article.

6. PLOS authors have the option to publish the peer review history of their article (what does this mean?). If published, this will include your full peer review and any attached files.

Reviewer #1: No

Reviewer #2: No

---

## [Author Response · Author response to Decision Letter 0]

19 Jun 2022

A rebuttal letter that responds to each point raised by the academic editor and reviewer(s) has been uploaded under a file labeled 'Response to Reviewers'.

---

## [Decision Letter · Decision Letter 1]

19 Jul 2022

Prioritized and Predictive Intelligence of Things Enabled Waste Management Model in Smart and Sustainable Environment

PONE-D-22-03543R1

Dear Dr. Gaber,

We’re pleased to inform you that your manuscript has been judged scientifically suitable for publication and will be formally accepted for publication once it meets all outstanding technical requirements.

Kind regards,

Chakchai So-In, Ph.D.

Academic Editor

PLOS ONE

Additional Editor Comments (optional):

Reviewers' comments:

Reviewer's Responses to Questions

**Comments to the Author**

1. If the authors have adequately addressed your comments raised in a previous round of review and you feel that this manuscript is now acceptable for publication, you may indicate that here to bypass the “Comments to the Author” section, enter your conflict of interest statement in the “Confidential to Editor” section, and submit your "Accept" recommendation.

Reviewer #1: All comments have been addressed

Reviewer #2: All comments have been addressed

2. Is the manuscript technically sound, and do the data support the conclusions?

Reviewer #1: Yes

Reviewer #2: Yes

3. Has the statistical analysis been performed appropriately and rigorously? 

Reviewer #1: Yes

Reviewer #2: Yes

4. Have the authors made all data underlying the findings in their manuscript fully available?

Reviewer #1: Yes

Reviewer #2: Yes

5. Is the manuscript presented in an intelligible fashion and written in standard English?

Reviewer #1: Yes

Reviewer #2: Yes

6. Review Comments to the Author

Reviewer #1: I consider the authors sufficiently addressed the suggestions and comments. The paper is now good for publication.

Reviewer #2: In this revision, all concerns are revised well . Overall, I believe that this research is interesting and useful, the related work, presentation, and experimental results are mature enough to merit for publication in this journal. Therefore, I suggest that this paper can be accepted

7. PLOS authors have the option to publish the peer review history of their article (what does this mean?). If published, this will include your full peer review and any attached files.

Reviewer #1: No

Reviewer #2: No

---

## [Editor Report · Acceptance letter]

25 Jul 2022

PONE-D-22-03543R1 

Prioritized and Predictive Intelligence of Things Enabled Waste Management Model in Smart and Sustainable Environment 

Dear Dr. Gaber:

I'm pleased to inform you that your manuscript has been deemed suitable for publication in PLOS ONE. Congratulations! Your manuscript is now with our production department. 

Kind regards, 

on behalf of

Dr. Chakchai So-In 

Academic Editor

PLOS ONE